# Phospholipids, the Masters in the Shadows during Healing after Acute Myocardial Infarction

**DOI:** 10.3390/ijms24098360

**Published:** 2023-05-06

**Authors:** Dan-Valentin Pistritu, Anisia-Cristiana Vasiliniuc, Anda Vasiliu, Elena-Florentina Visinescu, Ioana-Elena Visoiu, Smaranda Vizdei, Paula Martínez Anghel, Antoanela Tanca, Octavian Bucur, Elisa Anamaria Liehn

**Affiliations:** 1Victor Babes’ National Institute of Pathology, 99-101 Splaiul Independentei, 050096 Bucharest, Romania; 2Faculty of Human Medicine, Carol Davila University of Medicine and Pharmacy, 37 Dionisie Lupu Street, 020021 Bucharest, Romania; 3Business Academy Aarhus, 30 Sønderhøj, 8260 Viby J, Denmark; 4Viron Molecular Medicine Institute, 201 Washington Street, Boston, MA 02108, USA; 5Institute for Molecular Medicine, University of Southern Denmark, 25 J.B Winsløws Vej, 5230 Odense, Denmark; 6National Heart Center Singapore, 5 Hospital Dr., Singapore 169609, Singapore

**Keywords:** myocardial infarction, phospholipids, inflammation, scar formation, extracellular matrix, ventricular remodeling

## Abstract

Phospholipids are major components of cell membranes with complex structures, high heterogeneity and critical biological functions and have been used since ancient times to treat cardiovascular disease. Their importance and role were shadowed by the difficulty or incomplete available research methodology to study their biological presence and functionality. This review focuses on the current knowledge about the roles of phospholipids in the pathophysiology and therapy of cardiovascular diseases, which have been increasingly recognized. Used in singular formulation or in inclusive combinations with current drugs, phospholipids proved their positive and valuable effects not only in the protection of myocardial tissue, inflammation and fibrosis but also in angiogenesis, coagulation or cardiac regeneration more frequently in animal models as well as in human pathology. Thus, while mainly neglected by the scientific community, phospholipids present negligible side effects and could represent an ideal target for future therapeutic strategies in healing myocardial infarction. Acknowledging and understanding their mechanisms of action could offer a new perspective into novel therapeutic strategies for patients suffering an acute myocardial infarction, reducing the burden and improving the general social and economic outcome.

## 1. Cardiovascular Diseases: The Unanswered Problem of the Century

Cardiovascular diseases are the leading cause of death worldwide. Almost half of cardiovascular diseases are represented by coronary heart disease [1], with its most dramatic and destructive complication: acute myocardial infarction [2].

Acute myocardial infarction is the result of a sudden decrease in coronary blood flow, which induces an imbalance in the oxygen consumption in cardiomyocytes. Independent of the cause of the unsatisfactory coronary irrigation, the sudden reduction of oxygen induces permanent damage to the heart cells, such as apoptosis or ischemic necrosis. As a consequence, the heart function will be affected along with the increased proneness to arrhythmias [2,3]. Reperfusion of the blocked vessel minimizes damages; however, it does not always succeed in preventing subsequent ventricular remodeling and heart failure [2]. 

Despite extensive research in the last decades, besides the revascularization interventions, we have limited therapeutic strategies which can be involved in the treatment of patients suffering from acute myocardial infarction [4,5]. Therefore, in this review, we would like to focus on phospholipids, a particular class of biological compounds with essential cellular functions. Alteration in the phospholipid profile is always associated with increased cardiovascular disease occurrence. Reduced phospholipids content in the HDL particle is associated with a high incidence of cardiovascular diseases [6] and increased oxidation of phospholipids, being an excellent biomarker for the early detection of oxidative stress associated with myocardial diseases [7].

Phospholipids were intensively studied in the past. However, since the rushed development of novel methods for studying proteins such as immunobiological targeting with antibodies or mRNA/DNA detection by PCR techniques, their importance and role were shadowed by the difficulty or incomplete available research methodology to study their biological presence and functionality. Thus, while mainly neglected by the scientific community, phospholipids could represent an ideal target for future therapeutic strategies in healing myocardial infarction. 

## 2. Phospholipids: Old Knowledge Rediscovered

Phospholipids are major structural components of cell membranes; they are amphiphilic molecules consisting of two hydrophobic tails represented by fatty acid chains and one hydrophilic head composed of a phosphate group linked to an alcohol (glycerol or sphingosine). Glycerol-based phospholipids (glycerophospholipids) possess a varying phosphate group (phosphatidic acid), which can be modified with different molecules such as choline (phosphatidylcholine), ethanol (phosphatidylethanolamine), serine (phosphatidylserine) and inositol (phosphatidylinositol) [8]. The other class of phospholipids are sphingosine-based PL (sphingolipids) and include species such as sphingosine-1-phosphate (S1P), sphingomyelin (SM) and ceramides (Cer).

Phospholipids also constitute the main components of the sarcolemma in cardiomyocytes, where phosphatidylcholine (PC) represents 45% of the membrane content and phosphatidylethanolamine (PE) represents 37%, respectively. Fewer amounts of other phospholipids are found in isolated Sprague Dawley rat hearts in different concentrations, such as sphingomyelin (SM), phosphatidylinositol (PI), phosphatidylserine (PS) and phosphatidylglycerol (PG) [9]. The main knowledge about different groups of phospholipids, as known so far, is summarized in Table 1. 

**Phosphatidylcholine (PC)** represents almost 50% of total cellular phospholipids, present mostly on the outer surface of the bilayer membrane, and plays structural and functional roles in the cell membrane [10]. PC is synthesized in the liver, where it also participates in the generation of plasma lipoproteins such as HDL (high-density lipoprotein) and VLDL (very low-density lipoprotein) [11]. In the lungs, PC and dipalmitoylphosphatidylcholine (DPPC) make up to 85% of the lipid component of the phospholipid-rich surfactant, providing the surfactant with its surface tension reduction properties [12,13]. In the colon epithelia, PC from mucus protects against bacterial infections [14]; thus, altered PC concentrations in the intestinal mucus have been evidenced as a possible cause of ulcerative colitis [15]. Moreover, a single intraperitoneal administration of 75 mg/kg of PC for a 30 min duration inhibits bacterial adherence/growth, as well as peritoneal carcinosis, by inhibiting intraperitoneal adhesion of tumor cells accomplished by a 150 mg/kg dose of phospholipids [50]. However, in cardiovascular diseases, dietary PC supplementation is linked to increased all-cause and cardiovascular mortality, notably in the diabetic population [16].

**Phosphatidic acid (PA) and its lysoform lysophosphatidic acid (LPA)** are bioactive phospholipids, involved in the regulation of various cellular responses, by acting on its specific G-protein coupled transmembrane receptors. PA is mainly formed by hydrolysis of phosphatidylcholine after the activation of phospholipase D (PLD). LPA is secreted by several cell types, such as platelets, fibroblasts, adipocytes and cancer cells and can be found in several biological fluids, such as serum, plasma and aqueous humor [17]. LPA is involved in many chronic inflammatory diseases [51] and cancers [52]. In mice, LPA is increased in peripheral blood in the early stages after MI, while its receptor LPA2 expression increases in the heart, promoting angiogenesis [18], and also in in vitro stimulation of myocardial cell proliferation through LPAR3 [19]. 

**Phosphatidylethanolamine (PE)** is a major structural lipid located predominantly on the inner leaflet of membranes, including the mitochondrial membrane. Its distribution is heterogeneous between different organs, from 45% in the brain to 20% in the liver [53]. PE is involved in different pathologies of the brain, liver and skeletal muscle [54]. It seems that PE modulates the insulin sensitivity of the muscle through a cellular calcium homeostasis mechanism, since reduced PE levels in the sarcoplasmic reticulum are correlated with suppressed activity of SERCA [10]. Moreover, reduced levels of PE correlate with α-synuclein accumulation and the formation of Lewy bodies in Parkinson’s disease [55], which can be corrected by PE dietary supplementation in yeast and Caenorhabditis elegans worm models of Parkinson’s disease [20]. There are few data regarding the involvement of PE in cardiovascular diseases; however, it seems that PE biosynthesis increases during the differentiation of P19 teratocarcinoma cells into cardiomyocytes [21].

**Phosphatidylserine (PS)** is located in the inner layer of the plasma membrane, involved in apoptosis signaling by its transition to the outer leaflet of the membrane [22]. 

Even though PS is not so abundant compared to other phospholipids, its biological significance is much more relevant, since it participates in intracellular signaling, activating protein kinases C [23,24]. Thus, PS is involved in blood coagulation [56], memory and cognitive function through modulation of neurotransmitter release [57], viral infection of macrophages [58], hypothalamo-pituitary-adrenal axis modulation with reduction of cortisol levels and enhancement of sport performance [25,59]. PS is also useful in imaging diagnosis by synthesizing PS-rich microbubbles to be tracked through ultrasound techniques [60,61,62] in order to assess inflammation in experimental ischemic conditions such as brief (10 min) myocardial ischemia in wild-type C57Bl/6 mice [26]. 

**Phosphatidylinositol (PI)** is also involved in intracellular signaling. PI phosphorylation generates various products and species such as PIP2 (phosphatidylinositol 4,5 bisphosphate), which are involved in cellular functions, including signaling, membrane traffic, and ion channel regulation and actin cytoskeleton architecture and dynamics [63]. In cardiomyocytes, PIP2 depletion not only results in T-tubule deprivation and impaired Ca^2+^ handling [27] but also hypertrophy, heart failure and diabetic cardiomyopathy [27,28,29]. PI mediates the interaction with different target cytosolic proteins, regulating their membrane translocation and vesicular trafficking, while serving as a precursor for other signaling lipids, including Toll-like receptor signaling and trafficking, with crucial effects on the innate immune response [64]. Further, PI hydrolysis by phospholipase C generates a well-known and biologically active metabolite, **DAG (diacylglycerol)**. DAG is not only a basic membrane component but also a central element in lipid-mediated signaling [65]. DAG dysregulations result in impaired organ and cell development and have been associated with numerous pathologies such as diabetes, immune system disorders, cancer and Alzheimer’s disease [66]. In heart disease, DAG accumulation has been linked to cardiac hypertrophy in a mice study, but the underlying mechanisms are still unknown [30].

**Cardiolipin (CL)** is a form of diphosphatidylglycerol located exclusively in the inner membrane of mitochondria, where it constitutes about 20% of the total lipid composition, activating numerous enzymes involved in mitochondrial bioenergetic metabolism [31]. Alterations in CL biosynthesis and homeostasis have been linked to numerous inherited disorders, such as Senger syndrome [32], MEGDEL syndrome [33], neurological diseases with frontotemporal Dementia [37], renal microvascular rarefaction [34] or nonalcoholic fatty liver disease [35]. In the heart, ischemia/reperfusion injury and heart failure are associated with significant changes in the cardiolipin mitochondrial content [36]. 

**Sphingomyelin (SM)** is one of the main sphingolipids located in the membrane that is involved in cell apoptosis and cell migration. SM supports brain myelination and helps regulate chromatin function; therefore, alteration of SM can cause severe neurological diseases [38]. In the heart, SM was linked to an increased probability of developing coronary heart disease [39] and heart failure [40]. Sphingomyelin phosphodiesterase 1 (SMPD1) is the key enzyme that produces about 70% of total cellular ceramides in the heart, converting sphingomyelins to the pro-inflammatory and pro-apoptotic second messenger ceramides [67]. 

**Ceramides (Cer)** are another group of sphingolipids, which have a lesser structural role, but interfere with numerous basic physiological functions of the cells, such as cell growth, differentiation, senescence and apoptosis [41,42,43]. Similar to other sphingolipids, Cer are important in numerous pathologies, such as cancer [68], neurodegeneration [69], inflammatory/autoimmune diseases [70,71], diabetes [72] and atherosclerosis [73,74]. Besides their role in atherosclerosis and valvular diseases [44], Cer levels were found to be significantly increased in patients with left ventricle chronic ischemia [75], correlating with plaque rupture and the severity of coronary artery stenosis in patients with acute myocardial infarction [45]. 

**Sphingosine-1-phosphate (S1P)** is a sphingolipid which, by interaction with its specific receptors (S1P Receptor 1–5, high-affinity G-protein-coupled receptors), regulates multiple mechanisms of cells and tissues [46,47,49], such as proliferation, migration, morphogenesis, inflammation and angiogenesis [76]. S1P is involved in different pathologies, such as immune diseases [46], atherosclerosis [48], liver oxidative injury and fibrogenesis [49], kidney fibrosis in the case of chronic kidney disease [77] or osteoporosis [47]. Moreover, targeting the S1P Receptor using FTY720, an immunomodulatory drug, was the very first orally administered treatment accepted for relapsing-remitting multiple sclerosis [46].

Despite the multitude of experimental data demonstrating the importance and involvement of phospholipids in cardiovascular diseases, there is currently only very limited information highlighting the benefits of phospholipids in clinical settings [78]. The simple and multiple formulations, production and synthesis, low prices and high availability due to the known minimal side effects proved to have a double-edged sword effect, making them unattractive for the big pharma investors, who do not recognize them as a possible profit and economic gain. To counteract this issue, many centers specialized in studying the effects of phospholipids were financed with public resources and are currently trying to overcome all these challenges [79] and restore the role of phospholipids as an important therapeutic target, particularly for cardiovascular diseases.

## 3. Healing after Myocardial Infarction: The Unsolved Puzzle of the Heart

Acute myocardial infarction is the most feared complication of coronary diseases, with severe consequences not only on individual health and well-being but also regarding the social and economic burden of the community. The healing process after myocardial infarction was studied intensively [80,81]; however, no novel therapies have been developed to improve ventricular remodeling and preserve heart function. Since we are far from understanding the complexity of the processes taking place in the heart during myocardial infarction [80,81], it is difficult to design novel therapies that can pass the experimental phase and prove efficiency in clinical settings [82]. On top of that, there are heterogeneous methodologies and experimental conditions all over the world, which question reproducibility and make result interpretation difficult [83]. 

Briefly, healing after myocardial infarction can be divided into three overlapping phases, with different characteristics and outcomes [84]. The inflammatory phase is the initial phase starting immediately after an ischemic event in the heart [85], characterized by rapid neutrophils infiltration [86,87] and the creation of an inflammatory environment [87,88] to allow the important players to enter the injured site, such as the monocytes or fibroblasts, and prepare for the consequent phases. Neutrophils are the most controversial cells [89,90], initially thought of as a potential therapeutic target [91], but with detrimental results in clinical application [92], which highlighted the importance of knowing the detailed mechanisms and biological processes before any clinical attempt. Neutrophil infiltration induces shortly after myocardial infarction a massive upregulation of all kinds of cytokines and chemokines, so called “cytokine storm”, which correlates directly with the size of the affected areas, leading to additional cardiomyocyte apoptosis, thus worsening the prognosis [93]. The proliferation phase overlaps with the inflammatory phase and produces tissue as a mechanical support for the injured heart after the cellular debris is removed. Fibroblasts and monocytes later synthesize TGF-β1 [89,94], which will stop the inflammatory processes [95] and allow the healing phase to continue and fulfill the mature scar. Phospholipids are actively involved in all the processes and events during healing after myocardial infarction, as summarized in Figure 1. 

### 3.1. Phospholipids—The Quiet Leader behind the Doors

The role of phospholipids was recognized a long time ago, since their level was found to be changed in the plasma of patients with coronary artery disease [96], having the same prediction capacity as the other lipids [97]. Particularly, the decrease in phospholipids’ plasma levels was accompanied by increased plasma levels of lysophospholipids and oxidized phospholipids after ischemia/reperfusion injury [97,98]. Similar results were found in the ischemic myocardium, with decreased phospholipids and increased lysophospholipids in the myocardial tissue. 

Interestingly, this difference is more pronounced after the ischemia/reperfusion procedure, compared to chronic ischemia in an ex vivo rat model [99]. The detrimental effects of reperfusion on phospholipids are attributed to the presence of excessive reactive oxygen species (ROS). Since phospholipids are vulnerable to oxidation [100], their interaction with ROS results in the building of hydroperoxides, which are highly reactive molecules involved in cell death mechanisms [101]. Oxidized phospholipids are important in many pathologies [102,103,104], some of them being also valuable as biomarkers [103,105]. For a short period of ischemia, our body has resources for protection against oxidation, such as phospholipase A2 (PLA2) [106]. However, more than 3 h of myocardial ischemia followed by reperfusion induces an approximate 15% depletion of the phospholipid’s concentration in the heart in a model of experimental myocardial infarction in pigs [97]. The lowering of the level of phospholipids leads to membrane instability and electrophysiological changes which could translate into life-threatening ventricular arrhythmias [107]. Moreover, dysregulations of CL within the mitochondria can result in mitochondrial dysfunctions, with reduced ATP production and impaired bioenergetic metabolism [108,109]. In parallel, the increase in lysophospholipids, particularly Lysophosphocoline (LPC) [106], interferes with the ion currents in the membrane [110] and predisposes to dangerous ventricular arrhythmias [107,111,112]. 

**Phosphatidylcholine (PC)** is one class of myocardial phospholipids that is especially prone to oxidation [113]. Fragmented OxPCs such as POVPC [1-palmitoyl-2-(5′-oxo-valeroyl)-sn-glycero-3-phosphocholine], SOVPC [1-stearoyl-2-(5′-oxo-valeroyl)-sn-glycero-3-phospho-choline], SONPC [1-stearoyl-2-(9-oxo-nonanoyl)-sn-glycero-3-phosphocholine], PONPC [1-palmitoyl-2-(9-oxo-nonanoyl)-sn-glycero-3-phosphocholine] are elevated in the plasma of STEMI patients and are of interest due to their increased bioactivity [114]. Their reduction in experimental studies reduces the infarction size and ventricular remodeling [115]. Interestingly, OxPC appears to activate neutrophils through the receptor for platelet-activating factor [116]. Additionally, exposure of cardiomyocytes to aldo-OxPCs (POVPC and PONPC) resulted in ferroptotic death [117]. Ferroptosis, a novel mechanism of cell death involving phospholipid peroxidation, is an iron-dependent regulated cell death (RCD) mechanism associated with the accumulation of oxidized phospholipids [118] and mitochondrial destruction [119]. 

On the other hand, partial hydrolysis of phosphatidylcholine induces the formation of **Lysophosphatidylcholine (LPC)**. LPC is increasingly recognized as an important biomarker of cardiovascular diseases, such as myocardial infarction, atherosclerosis or diabetes [120]. LPC is produced by apoptotic cells and neutrophils, inducing the chemotaxis of monocytes and macrophages, which present specific receptors for LPC [121]. 

Other forms of PC, such as polyenylPC or dilinoleoylPC, are described to have an important anti-fibrotic role in the cleavage of collagen in hepatic cells [122]; however, there are no data regarding the anti-fibrotic activity in the heart during healing after myocardial infarction. 

**Phosphatidic acid (PA)**, mainly formed by the hydrolysis of phosphatidylcholine, increases the intracellular concentration of free Ca^2+^ in adult cardiomyocytes and augments the cardiac contractile activity of the heart [123,124]. Almost all positive inotropic agents increase the level of PA in cardiac sarcolemma [123]. PA also stimulates protein synthesis in cardiomyocytes, by enhancing the activity of PLC and protein kinase C [125,126,127]. Further, binding of PA to α1-microglobulin activates Akt, NF-kB and ERK signaling in the infarcted and border areas, stimulating processes such as inflammation, macrophage migration and polarization, while inhibiting fibrogenesis [128]. **Lysophosphatidic acid (LPA)** and its receptors promote cardiac regeneration and function through signaling pathways such as PI3K/AKT, BMP-Smad1/5, Hippo/YAP and MAPK/ERK [19]. Moreover, LPA reduces fibrosis and ventricular remodeling after myocardial infarction [18], increasing angiogenesis and endothelial cell proliferation and functionality through PI3K-Akt/PLC-Raf1-Erk and PKD1-CD36 pathways [18]. 

**Phosphatidylethanolamine (PE)** is involved in inflammatory processes after myocardial infarction by binding to α1-microglobulin, which seems to increase the mRNA expression of inflammatory cytokines and chemokines, decreasing α-smooth muscle actin and collagen 3a1 [128]. PE is found in blood in a range of 0–12 μM and in breast milk in a range of 46 μM [1]. It was also believed that PE can induce ferroptosis by its oxidation in the endoplasmic reticulum compartment [102]. However, PE plays a significant role in protein synthesis as a lipid chaperone and also in triggering autophagy in human studies, processes that are of great importance post-myocardial infarction [129]. It also increases the resistance to oxidative stress, as highlighted in a study on *Caenorhabditis elegans* [130], and seems to contribute to uncoupling protein 1-dependent respiration without compromising electron transfer efficiency or ATP synthesis [131]. However, its role is still far from being completely understood. 

**Phosphatidylserine (PS)** levels are decreased by 33% in the plasma of patients with myocardial infarction [98]. Besides the role of PS in cardioprotection [132,133], PS has a strong anti-inflammatory activity, reducing neutrophil activation [132]. PS is known to have a significant role in protection against diabetes, an important risk factor for myocardial infarction, by not only controlling insulin secretion, signaling and transduction but also mediating coagulation disorders in the microvasculature or targeting mitochondria [134]. A study overexpressing phosphatidylserine-specific phospholipase A1 was shown to reduce inflammation by inhibiting the phosphorylation of MAPKs such as p38, ERK and JNK, resulting in decreased TNF-α, IL-1β and NO secretion in M1 macrophages [135]. The administration of PS-containing liposomes has been revealed to protect against type 1 and type 2 diabetes in animal studies [134] and to also modulate the monocyte phenotype [136]. Furthermore, PS are prone to oxidation processes. However, unlike PC, the oxidized form of PS (OxPS) has beneficial effects, inhibiting macrophage production of nitric oxide (NO) and IL-1β transcription in a dose-dependent manner [137]. 

**Phosphatidylinositol (PI)** has been documented to participate in the ischemic preconditioning signaling pathway [105], initiating almost all cardioprotective pathways. It is the most studied phospholipid, acting as a substrate for the PI-3-kinase intracellular signaling pathway [138]. Thus, PI plays an important role in the development of different types of cardiomyopathy [27,28,29] but is also the main promotor of angiogenesis in the infarcted heart [139]. PI turnover seems to correlate with myocyte hypertrophy and increased performance, being significantly increased by angiotensin II one week after myocardial infarction. The PI metabolite, **DAG** (**diacylglycerol**), also plays an important role in left ventricular remodeling, being involved in post-myocardial infarction dysfunction and mortality [140]. As a co-activator of protein kinase C (PCK) in the cardiac cell, DAG interferes with preconditioning pathways, a well-known mechanism of cardioprotection [141], enhancing the tolerance to ischemia/reperfusion injury [142]. 

**Cardiolipin (CL)**, a mitochondrial phospholipid, plays an important role in cardiac homeostasis. Cardiolipin is particularly vulnerable to reactive oxygen species (ROS), leading to alterations in cardiolipin structure, particularly in ischemic conditions [143]. The alterations in cardiolipin are responsible for mitochondrial dysfunction, increased ROS production and apoptosis [144,145] observed during myocardial ischemia after myocardial infarction. 

**Sphingomyelin (SM)** is less studied in the pathophysiology of myocardial infarction compared to its derivates ceramides and sphingosine-1-phosphate. However, mice deficient in SM exhibit moderate neonatal lethality, a decrease in insulin secretion, inflammatory signals and atherosclerosis, as well as a reduction in mitochondrial function and ROS accumulation [76]. 

**Ceramides (Cer)** play important roles in healing after myocardial infarction, as a precursor of S1P [146]. Besides its role as a precursor, Cer is a key player in apoptosis and autophagy although the exact mechanism by which this occurs is not fully understood [76]. Cer can also impair the respiratory chain and open the mitochondrial permeability transition pores, increasing ROS production in myocardial ischemia/reperfusion injury [147,148], while also contributing to the cytochrome c release. A decrease in Cer levels was observed in cardioprotection due to ischemic preconditioning [149]. Although Cer and S1P are members of the same phospholipid class, their biological effects are opposite. High levels of Cer are found in the post-infarcted human myocardium and are correlated with increased cell death [150] and fibrosis and worsening of heart function [151,152]. 

**Sphingosine-1-phosphate (S1P)** is an essential component of high-density lipoproteins [68], exerting cardioprotective effects by ERK-related pathways [153,154] and anti-inflammatory effects during healing after myocardial infarction in mice [155]. S1P is one of the main modulators of angiogenesis during scar formation, not only by controlling vascular tone, endothelial and smooth muscle cell proliferation [76,156,157] but also activating CXCR4 phosphorylation and Jak2 phosphorylation [157] through the S1PR1/PI3K/Akt pathway [158]. Together with apolipoprotein M, they improve endothelial homeostasis by activating G-protein-coupled receptors [153,159]. High levels of S1P enhanced bone-marrow-derived progenitor cells recruitment to the infarcted myocardium and reduced ventricular remodeling and infarction scar in mice [160]. 

There are 5 types of S1P receptors: sphingosine-1-phosphate receptor 1 (S1PR1), S1PR2, S1PR3, S1PR4 and S1PR5. Among these, only the first 3 can be found in cardiovascular tissue, with S1PR1 being the most abundant in cardiomyocytes while S1PR3 is the predominant one in cardiac fibroblasts [156]. Interestingly, S1PR1 deletion has a significant effect on reparative macrophage accumulation in later stages post-myocardial infarction [146]. Binding S1P to S1PR2 mediates recruitment of muse cells into infarcted areas, a type of non-tumorigenic endogenous pluripotent-like stem cells, with the ability to differentiate into cells expressing cardiac markers, thus reducing infarct size and improving heart function after engraftment into male rabbit hearts [161]. Binding of S1P to its receptor S1PR3 in fibroblasts leads to increased migration and proliferation, essential for fibrotic development and cardiac remodeling [156,162]. S1P triggers TGFβ1 signaling and α-smooth muscle actin in fibroblasts, modulating the production of collagen [156]. The role of phospholipids during healing after myocardial infarction is summarized for convenience in Table 2. 

### 3.2. Therapeutical Strategies: Are They Really Novel?

Despite the general belief, besides the reperfusion interventions, there are only limited drug classes which proved efficient and can be used during acute myocardial infarction to protect the heart and preserve its function, including beta-blockers and angiotensin-converting enzyme inhibitors. Since there are still no reliable treatment options to limit myocardial infarction and reperfusion injuries, the imperative need for further research to gain a better understanding of myocardial infarction and develop an effective treatment is obvious. A summary of existent therapeutical strategies involving phospholipids is presented in Figure 2. 

Phospholipids seem to have been used since ancient times to treat cardiovascular disease, although not in the context of the knowledge we currently have. Consumption of certain foods containing bioactive lipid molecules, such as phospholipids, was shown to have beneficial effects for patients suffering from heart disease, particularly nuts and legume seeds [163]. Oil palm fruit (*Elaeis guineensis*) containing high amounts of phospholipids was used to prevent cardiovascular diseases [164], although it was observed that the high-temperature processing increased phospholipids oxidation, with consecutive adverse effects on health (i.e., organo-toxicity to the heart) [165]. *Sinomenium acutum* is a plant containing a large variety of bioactive components, including alkaloids and phospholipids and is utilized in traditional Chinese medicine to reduce arrhythmias and to sustain cardiac health [166]. Recently, a traditional Chinese herbal formula interfering with the phospholipid peroxidation pathway, called the Danlou Tablet, has been demonstrated to have cardioprotective benefits and reduce pain in patients presenting angina pectoris [167]. 

Even novel developed drugs, such as imidapril (an angiotensin-converting enzyme inhibitor) was demonstrated in animal studies to prevent heart failure by indirectly modulating the levels of PA, PC, SL and DAG [127] and thus reducing fibrosis, collagen accumulation and ventricular remodeling [168]. 

Similarly, a novel formulation of the phospholipid-aspirin complex (PL-ASA) was designed to reduce local acute gastrointestinal injury as well as increase platelet inhibition compared to plain aspirin or enteric-coated tablets [169].

Nevertheless, soy is probably the most used plant in ancient and current traditional medicine. Soy extract can modulate the LDL/HDL ratio which is a strong predictor of cardiac health [170]. Moreover, the addition of soybean phospholipids to the lipid-lowering medication of patients with low-lipid-diet-resistant coronary artery disease and hypercholesterolemia not only lowers the cholesterol levels but also inhibits platelet aggregation as a secondary benefit [78]. Soybean contains PC, PE, PI and PS in similar amounts.

**Supplementation with soybean PC** was shown to reduce the total cholesterol, LDL, ApoA1 and fibrinogen in plasma, thus showing its beneficial properties in patients suffering from hypertension and obesity [78]. In patients suffering from type IIA hypercholesterolemia, the administration of soybean PC reduced the atherosclerotic risk by increasing the levels of ApoA1 and reducing the levels of ApoA2, ApoE and homocysteine [78]. Furthermore, the administration of a synthetic HDL, carrying a recombinant apolipoprotein A-IMilano and 1-palmitoyl-2-oleoyl phosphatidylcholine (POPC) protects the isolated rabbit heart from reperfusion injury and significantly reduces fibrosis, decreasing scar size [171]. Interestingly, since ulcerative colitis is considered a risk factor linked to cardiovascular diseases such as accelerated atherosclerosis, atrial fibrillation and heart failure [172,173], using PC enriched enteric lecithin to improve remission rates [174] can decrease corticosteroid dependence [175] and thus reduce stress associated cardiovascular events.

Another traditional herbal medicine intensively used to reduce the incidence of cardiovascular diseases is Ginseng [176]. Studies have revealed **LPA** as the actual active component binding to Gintonin in the composition of Ginseng [177]. In other studies, the administration of an LPA receptor 2 agonist reduced scar formation and improved vascular neoangiogenesis and heart function in experimental studies [18]. Similarly, inhibiting the autotaxin/LPA signaling pathway results in increased angiogenesis, reduced scar size and improved heart function recovery [178]. Activation of LPA receptor 3 also improves the proliferation of myocardial cells in vitro through intracellular signaling (PI3K/AKT, BMP-Smad1/5, Hippo/YAP and MAPK/ERK) [19]. 

**Dietary supplementation with PE** increases resistance to oxidative stress, as highlighted in a study on *Caenorhabditis elegans* [130]. However, PE has emerged as a useful molecule for diagnostic procedures. It seems that a PE-based apoptotic tracer has the same efficiency as the Annexin-V tracer, with the additional benefit that radiation to non-target organs was lessened [179]. Moreover, polyethylene glycol (PEG) conjugated PE micelles demonstrate a prolonged resistance in the bloodstream and a high accumulation rate in the infarcted area of the heart in the rabbit model, making them a promising option for diagnosis and delivery in infarction medicine [180]. 

Phospholipid liposomes have been used more and more in the last period as delivery systems for different drugs to infarcted areas, including RNA-based therapeutics (such as siRNA, miRNA) [79,181]. 

**Oral administration of PS** is used largely to sustain skeletal muscle activity by athletes and during intense physical exercise [59] and cognitive activity of the brain [57]. It was also shown to decrease the post-infarction scar by 30% and offer cardioprotective benefits in a mouse model through the upregulation of protein kinase C-ε [132]. Moreover, PS administration significantly reduces the risk of developing diabetes, the main risk factor for cardiovascular disease. 

Further, **oral administration of DAG** can induce protection against injury during ischemia/ reperfusion in the rat heart [182]. Similarly, the administration of a selective inhibitor of DAG kinase-α reduces adverse LV remodeling, macrophage infiltration, inflammation and matrix metalloproteinase activity [128]. Moreover, the DAG metabolite 1-[1-11C]butyryl-2-palmitoyl-rac-glycerol (11C-DAG) has potential in imaging due to its high accumulation rate assessed by PET in regions with active phosphoinositide cycles, such as the ischemic myocardium in patients with anteroseptal MI [183].

Additionally, **cardiolipin (CL)** represents a reliable therapeutic target to sustain healing after myocardial infarction. 4-hydroxy-trans-2-nonenal, a signaling molecule formed from CL, exhibits a variety of biological activities, including inhibition of protein and DNA synthesis and activation of mitochondrial enzymes [144]. Even the protective effect of melatonin on reducing ROS formation is explained partly by its ability to preserve CL integrity [184]. Restoring mitochondrial health in an animal model of myocardial infarction seems to be possible by using pharmacological inhibition of Acyl-CoA lysocardiolipin acyltransferase 1 (known to catalyze acylation of lysocardiolipin back to cardiolipin) through the administration of Dafaglitapin [145]. 

Further, interference with **sphingolipids** by various sphingomyelinase inhibitors, such as 0.1 mg/kg intravenous of the sphingomyelinase blocker tricyclodecan 9 y-xanthate, is reported to decrease ceramide accumulation induced by ischemia/reperfusion injury and to reduce cardiomyocyte apoptosis in male New Zealand White rabbits [149]. Amitriptyline and desipramine are examples of sphingomyelinase inhibitors which have the ability to diffuse through membranes and enter various organelles. As a result, they hinder the binding of acid sphingomyelinase, which leads to the detachment and subsequent inactivation of the enzyme through degradation [185]. Targeting Sphingomyelin Phosphodiesterase 1 through Astaxanthin administration may have beneficial effects on cardiac health. Administering Astaxanthin, a natural carotenoid known for its potent antioxidant and anti-inflammatory properties, to mice after aortic constriction [186] or for a period of 4 weeks after undergoing MI induction surgery resulted in lowered collagen I/III ratio by decreasing phosphorylation and deacetylation of receptor-activated SMADs [187]. S1P receptors are another valuable therapeutic target. Using SEW2871 to pharmacologically activate S1PR1 notably reduces infarction size and increases myocardial contractility post-infarction in mice joined by parabiosis [146]. Acid Ceramidase modRNA also improves healing and cell survival after MI while enhancing cardioprotection [150]. Downregulation of serine palmitoyltransferase (an enzyme with a central role in the de novo formation of ceramides) or the excision of the gene responsible for its production leads to decreased scar size and pathological remodeling [152]. Similarly, myriocin administration proved to be effective in reducing the ceramide surge, suppressing markers of oxidative stress and inflammation (TNF-α, IL1-β and IL-6), thus reducing myocardial necrosis [188].

Phospholipids have minimal side effects due to their preexisting role as membrane components, thus being a class of compounds already familiar and abundant with the human body. Nevertheless, the oral administration, particularly of PC, was shown to have some adverse effects. The gut microbiota has the ability to metabolize PC to trimethylamine-N-oxide (TMAO), a molecule which has been linked to CVDs and increased mortality (especially in the diabetic population) [78]. Initially, the aforementioned health complications could be correlated only with TMAO alone but not with dietary PC intake. However, a study conducted in the US managed to illustrate a link between dietary consumption of PC and a higher probability of CVDs development and mortality (affecting more pronouncedly the diabetic demographic) [16]. Other side effects of PC, such as diarrhea, dizziness, nausea and intermenstrual bleeding, were reported by patients undergoing subcutaneous injections of PC for the purpose of subcutaneous fat reduction. These symptoms were described by approximately 3% of the studied patient cohort, while other milder, localized symptoms, such as pain, swelling and tenderness, were reported by a larger number of patients. The study surveying the risks of this procedure concluded that it had minimal risks when performed by experienced doctors, indicating that some of the side effects may have more to do with the method of administration and the human element [189].

In addition, PLs can even reduce the side effects of some drug classes, among which are NSAIDs. The administration of 100% native soybean PC for 14 days lessened the gastrointestinal side effects of NSAIDS such as gastroduodenal ulcer and abdominal pain [78]. In conclusion, the role of phospholipids in the therapy of cardiovascular diseases has been increasingly recognized in recent years. Used in singular formulation or in inclusive combinations with current drugs, phospholipids proved their positive and valuable effects in the protection of myocardial tissue, inflammation and fibrosis as well as regarding angiogenesis, coagulation or cardiac regeneration not only in animal models but also in human pathology.

### 3.3. Future Perspective: Simple Is Complicated

There are more and more valuable scientific data that demonstrate that phospholipids administration represents an inexpensive strategy to promote cardiomyocyte survival and the modulation of the inflammatory response after an acute myocardial infarction. Unfortunately, planning experimental studies or human clinical studies proved to be a real challenge for the scientific community. Administration of phospholipids in animal studies is simple, since their environment and food are strictly controlled and monitored. However, the analysis of their cycle in the organism, as well as their accumulation or interaction, is limited by the existent limited laboratory methods for lipid detection. The gold standard method for lipid detection and identification is chromatography [190]. Currently, additional methods are being developed which offer a better understanding of lipid biology, such as mass spectrometry [191,192], or matrix-assisted laser desorption/ionization (MALDI) time-of-flight (TOF) imaging mass spectrometry (IMS) [193]. While the methods provide detailed information about the lipid species and mass, allowing lipid identification, localization and interaction, they are very expensive and require highly trained experts for data acquisition and data processing. Thus, there are only a few laboratories in the world that can allow and afford this kind of research. Moreover, since no antibody against lipids can be synthesized, constructing therapeutic agents to target lipids and lipid pathways represents a huge challenge. We can speculate that lipid research requires new experimental approaches to revolutionize and allow the flourishing of lipid biology and research, like immunohistology and polymerase chain reaction (PCR) for protein research. 

Another challenge in phospholipid research is represented by human clinical study design [194]. Since phospholipids are found in all plants and animals, it is difficult to control and monitor phospholipid intake and its effect on a particular disease, since they are normally found in our food [195,196]. Particularly, fish and soy, which are consumed regularly as food all over the world, contain massive amounts of phospholipids and cannot be excluded from the diet during a long-term clinical study. Probably, the consumed food offers many of us already protection against diseases [197] and particularly against cardiovascular disease [198], and thus, we can argue for a different response to the risk factors and the surrounding environment depending on the phospholipid content of the ingested food. 

## 4. Conclusions: Too Simple to Be True

Since the occurrence of a myocardial infarction implies many structural and functional changes to the heart that promote and sustain the development of heart failure, it is of the utmost importance to minimize these alterations and prevent undesirable remodeling. Phospholipids demonstrate valuable functions and roles during healing after myocardial infarction, with important influence on all biological processes involved: inflammation, proliferation, angiogenesis and fibrosis. They have been used as therapeutic targets since ancient times; however, we have barely started to recognize and understand their complex role in the modulation and regulation of biological processes. The missing or negligible side effects of phospholipids administration, as known until now, make them ideal as therapeutic agents. Thus, acknowledging and understanding their mechanisms of action could offer a new perspective into the novel therapeutic strategies for patients suffering an acute myocardial infarction, reducing the burden and improving the general social and economic outcome.

## Figures and Tables

**Figure 1 ijms-24-08360-f001:**
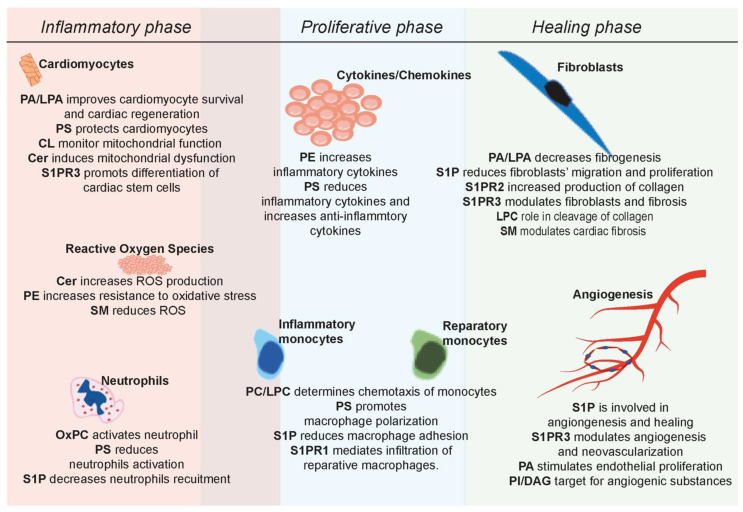
The role of phospholipids in healing after myocardial infarction. Phospholipids are involved in all phases after myocardial infarction, influencing inflammatory processes and recruitment of the immune cells, modulating the switch to the anti-inflammatory processes, fibrosis and angiogenesis, thus modulating the ventricular remodeling, scar formation and preservation of heart function after acute myocardial infarction (Cer—ceramides; CL—cardiolipin; DAG—diacylglycerol; LPA—lysophosphatidic acid; LPC—lysophosphatidylcholine; LPS—lysophosphatidylserine; PA—phosphatidic acid; PC—phosphatidylcholine; PE—phosphatidylethanolamine; PI—phosphatidylinositol; PKC—protein kinases C; PS—phosphatidylserine; ROS—reactive oxygen species; OxPC—oxidize phosphatidylcholine; OxPS—oxidize phosphatidylserine; S1P—sphingosine-1-phosphate; SM—sphingomyelin).

**Figure 2 ijms-24-08360-f002:**
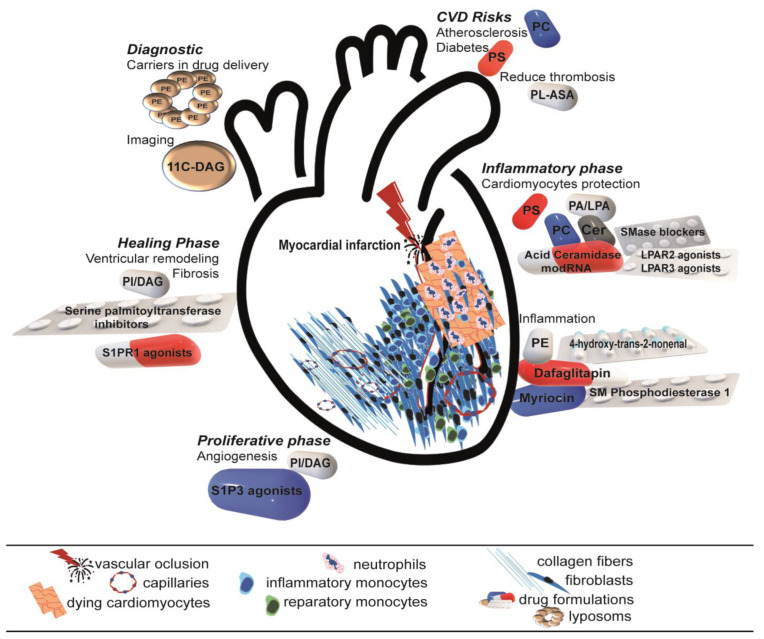
Therapeutical strategies involving phospholipids during healing after myocardial infarction. When administrated in different formulation, phospholipids modulate all phases after myocardial infarction. (CDV—cardiovascular diseases; Cer—ceramides; CL—cardiolipin; DAG—diacylglycerol; LPA—lysophosphatidic acid; LPC—lysophosphatidylcholine; LPS—lysophosphatidylserine; PA—phosphatidic acid; PC—phosphatidylcholine; PE—phosphatidylethanolamine; PI—phosphatidylinositol; PS—phosphatidylserine; S1P—sphingosine-1-phosphate; SM—sphingomyelin; ASA—acid acetylsalicylic).

**Table 1 ijms-24-08360-t001:** Localization and functions of phospholipids. HDL (high-density lipoprotein); VLDL (very-low-density lipoprotein); Lpar2-KO (lysophosphatidic acid receptor 2-knockout); ApoE-KO (apolipoprotein E-knockout); HPPCn (hepatopoietin Cn); FVB (friend leukemia virus B).

Phospholipid	Localization	Function	Experimental Design
PC	The outer surface of cell membrane	Membrane integrity	Human study [10]
	Liver	Generation of HDL, VLDL	Human study [11]
DPPC; PC	Lungs	Pulmonary surfactant component [12]	Ztm:MF1 mice [13]; Male Sprague Dawley rats [13]; Mixed-breed York-Pyatrain-Landrace pigs [13]; Human study [13]
PC	Colon	Protection against bacterial infections	Male rats [14]
	Colon	Possible cause of ulcerative colitis	Porcine lipid model [15]
	Cardiovascular system	Increases all-cause and cardiovascular mortality	Human study [16]
PA/LPA	Serum Plasma Aqueous humor	Intracellular signaling	Human and animal studies [17]
LPA	Heart	Suppression of fibrogenesis	(*Lpar2*-KO) MI mice model [18]
	Heart	Stimulation of myocardial cell proliferation	LPA3 and LPA1 knockout mice [19]; Neonatal Sprague Dawley rats [19]
PE	Skeletal muscle	Influences insulin sensitivity	Animal models [10]
	Brain	Reduces α-synuclein accumulation and the formation of Lewy bodies in Parkinson’s disease	Yeast; *Caenorhabditis elegans* worm models [20]
	Heart	Differentiation of P19 teratocarcinoma cells into cardiomyocytes	Cell culture [21]
PS	The inner layer of the plasma membrane	Apoptosis signaling	Cell culture [22]
	Cellular level	Activates protein kinases C	Cell culture [23,24]
	Skeletal muscle	Sports performance enhancement	Human study [25]
	Inner layer of plasma membrane	Inflammation assessment through ultrasound techniques	Wild-type C57Bl/6 mice [26]
PI	Cardiomyocytes	Regulation of T-tubules and Ca^2+^ handling	Isolated Sprague Dawley, Wistar Kyoto and spontaneously hypertensive rat hearts [27]; Isolated human hearts [27]
	Heart	Hypertrophy, heart failure and diabetic cardiomyopathy	Isolated Sprague Dawley, Wistar Kyoto and spontaneously hypertensive rat hearts [27]; Isolated human hearts [27]; Stroke-prone spontaneously hypertensive rats [28]; Wistar-Kyoto rats [28]; FVB/N male mice [29]
DAG	Heart	Linked to cardiac hypertrophy	Animal model [30]
CL	Inner membrane of the mitochondria	Mitochondrial bioenergetic metabolism	Human study [31,32,33,34,35,36]; Animal model [37]
	Heart	Development of inherited disorders	
		Involved in ischemia/reperfusion injury and heart failure	
SM	Brain	Myelination Regulation of the chromatin function	Experimental and human studies [38]
	Heart	Development of coronary heart disease	Human study [39]
		Development of heart failure	Human study [40]
Cer	Cellular level	Cell growth Cell differentiation Senescence Apoptosis	Experimental and human studies [41,42]; C57BL mice [43]
	Heart	Involved in the development of atherosclerosis and valvular diseases	*ApoE*-KO mice; murine model [44]
		Correlates with plaque rupture and the severity of coronary artery stenosis	Human study [45]
S1P	In plasma, transported by HDL and albumin	Intracellular signaling	Animal, cellular and human studies [46,47]
		Involved in atherosclerosis	Human study [48]
	Liver	Fibrogenesis	HPPCn^liver+/+^ transgenic FVB mice [49]

**Table 2 ijms-24-08360-t002:** The role of phospholipids in the heart. MI (myocardial infarction); *ALCAT1*-KO (Acyl-CoA:lysocardiolipin acyltransferase-1 knockout); CVDs (cardiovascular diseases); PLC (phospholipase C); ROS (reactive oxygen species); NO (nitric oxide); IL-1β (interleukin-1β); MAPK (mitogen-activated protein kinase); FVB/N (friend leukemia virus B/N).

Phospholipid	Function	Experimental Design
PC (OxPC)	Elevated in the plasma of STEMI patients	Human study [114]
	Associated with increased scar size and ventricular remodeling	Adult rat ventricular cardiomyocytes [115]
	Activates neutrophils	Human neutrophils [116]
	Induces ferroptosis	Adult rat ventricular cardiomyocytes [117]
LPC	Biomarker for CVDs (i.e., MI, atherosclerosis and diabetes)	Human study [120]
	Chemotaxis of monocytes and macrophages	Human and mouse monocytes, mouse model [121]
PA	Increases the intracellular concentration of free Ca2+ in adult cardiomyocytes; Auguments inotropism	Isolated rat cardiomyocytes, rat model [123]; Isolated Male Sprague Dawley rat cardiomyocytes [124]
	Stimulates protein synthesis in cardiomyocytes through augmentation of PLC and protein kinase C activity	Cell culture [125]; Male Sprague Dawley rats MI model [126,127]
	PA-α1-microglobulin complex stimulates inflammation, macrophage migration and polarization and inhibits fibrogeneiss in the infarct border area	Mouse MI model [128]
LPA	Encourages cardiac function	LPA3 and LPA1 knockout mice; neonatal Sprague Dawley rats [19]
	Lessens fibrosis and ventricular remodeling after MI	(*Lpar2*-KO) MI mice model [18]
	Increases angiogenesis and endothelial cell proliferation and functionality	
PE	PE-α1-microglobulin complex stimulates inflammationby increasing the mRNA expression of inflammatory cytokines and chemokines, decreasing α-smooth muscle actin and collagen 3a1	Mouse MI model [128]
	Induces ferroptosis	Cardiomyocyte cell culture [102]
	Protein synthesis as a lipid and chaperon	Cellular, plant and animal models and human studies [129]
	Triggers autophagy	
	Increases the resistance to oxidative stress	*Caenorhabditis elegans* worm models [130]
	Involved in uncoupling protein 1-dependent respiration without compromising electron transfer efficiency or ATP synthesis	Animal model [131]
PS	Cardioprotection	Mouse MI model [132,133]
	Reduces neutrophil activation	
	Protects against diabetes	Animal study [134]
	Anti-inflammatory activity by inhibiting phosphorylation of MAPKs	RAW264.7 macrophages culture [135]
PS-containing liposomes	Protects against type 1 and type 2 diabetes	Animal study [134]
	Modulate the monocyte phenotype	Mouse, rat and human cellular models, mouse, rat and pig myocardial I/R models, mouse and rat MI models, human studies [136]
OxPS	Inhibits macrophage production of NO and IL-1β transcription	RAW264.7 macrophage culture [137]
PI	Ischemic preconditioning	Human study [105]
	Cardioprotection	Transgenic mice [138]
	Development of different types of cardiomyopathies	Isolated Sprague Dawley, Wistar Kyoto and spontaneously hypertensive rat hearts [27]; Isolated human hearts [27]; Stroke-prone spontaneously hypertensive rats [28]; Wistar-Kyoto rats [28]; FVB/N male mice [29]
	Main promoter of angiogenesis in the infarcted heart	Various animal models (zebrafish, chicken embryos, mice) [139]
	PI turnover seems to correlate with myocyte hypertrophy and increased performance	Mouse MI model [140]
DAG	Left ventricular remodeling	Mouse MI model [140]
	Involved in post-myocardial infarction dysfunction and mortality	
	Preconditioning	Animal models; human myocytes [141]
	Enhances tolerance to ischemia/reperfusion injury	Transgenic mice ischemia model [142]
CL	Its alteration increases mitochondrial dysfunction, ROS production and apoptosis	Rat MI model [144] *ALCAT1*-KO MI mice model [145]
SM	Lowers rate of neonatal lethality	Sphingomyelin synthase (SMS)-KO mice; animal studies [76]
	Increases insulin secretion, inflammatory signals and atherosclerosis	
	Increases inflammatory signals	
	Protects against ROS accumulation and mitochondrial dysfunction	
Cer	Apoptosis and autophagy	Animal studies [76]
	Increases ROS production in myocardial ischemia/reperfusion injury	Mouse ischemia model [147,148]
	Lower levels following cardioprotection through ischemic preconditioning	Rabbit ischemia model [149]
	High levels of Cer in post-infarcted human myocardium	Mouse MI model [150]
	Increases cell death	
	Increases fibrosis and worsening of heart function post-MI	Human study [151,152]
S1P	Cardioprotective effects	Cellular and animal models, human studies [154]; Cellular and animal models (mouse, rat) [153]
	Anti-inflammatory effects during healing after myocardial infarction	In vivo mouse model of myocardial ischemi/reperfusion [155]
	Main modulator of angiogenesis during scar formation	Animal model [76]; Humans, human cells, mouse MI/reperfusion model, rat cardiomyocytes, isolated and perfused rat hearts [156]; patient-derived endothelial progenitor cells and mouse model of hind limb ischemia [157]
	Controls vascular tone, endothelial and smooth muscle cell proliferation	Animal model [76]; Humans, human cells, mouse MI/reperfusion model, rat cardiomyocytes, isolated and perfused rat hearts [156]; Patient-derived endothelial progenitor cells and mouse model of hind limb ischemia [157]
	Activates CXCR4 phosphorylation and Jak2 phosphorylation	Patient-derived endothelial progenitor cells and mouse model of hind limb ischemia [157]
	Improves endothelial homeostasis together with ApoM	Cellular and animal models, human studies [153]
	Enhances the recruitment of bone-marrow-derived progenitor cells to the infarcted myocardium	Mouse MI model [160]
	Reduces ventricular remodeling and infarction scar	
	Binding of S1P to S1PR1 affects reparative macrophage accumulation at later stages post-myocardial infarction	Mouse model of MI and parabiosis [146]
	Binding S1P to S1PR2 mediates recruitment of muse cells into the infarcted areas, reducing infarct size and improving heart function	Rabbit model of AMI and human and rabbit muse cells [161]
	Binding of S1P to S1PR3 in fibroblasts increases migration and proliferation	Humans, human cells, mouse MI/reperfusion model, rat cardiomyocytes, isolated and perfused rat hearts [162]; rat MI model [156,162]
	Modulates the production of collagen	Humans, human cells, mouse MI/reperfusion model, rat cardiomyocytes, isolated and perfused rat hearts [156]

## Data Availability

Not applicable.

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
