# Peer review of "Phospholipids, the Masters in the Shadows during Healing after Acute Myocardial Infarction"

_ijms, 2023, doi:10.3390/ijms24098360_

Round 1

Reviewer 1 Report

Despite my opinion transcription regarding what is requested in the electronic form from International Journal of Molecular Sciences MDPI, I would like to add the following comments:

The article is generally good and follows all the precepts required by International Journal of Molecular Sciences MDPI and is therefore in accordance with its publication standards.

The content of the article is interesting, correct and important in the context of acute myocardial infarction. In fact, it is a scientific area that has been forgotten with regard to the area of acute myocardial infarction and the shadows that exist regarding its evolution and the methodology to be used to control and minimize its deleterious effects on the individual who suffers it; The data described point to a window of therapeutic opportunities that this pathology, so frequent and so important, demands of all of us.

It is a narrative review with 3 well-structured chapters with scientifically correct content and appropriate citations. However, overall it has three aspects that should be improved and that will be mentioned later: there should be an indication throughout the text, whenever possible, of the experimental animal/human species model in which the referenced results were obtained; the plasma concentrations obtained (at least in some studies) of the phospholipids described, and the dose of some phospholipids used orally; incidentally it may be important to construct a Table [as for example with the topics: Reference No. of participants Study design Results (intervention group versus control) P values), or of another type]; there is still one or another inaccuracy to be noted which will be properly referenced below.

So, and throughout the "article topics" (in red purposeals to be potentially changed)

Title/authors: 3Business Academy Aarhus, Denmark was missing.

Abstract

- "This review focus on the current knowledge about the role of phospholipids in the pathophysiology (missing) and therapy of cardiovascular diseases"

1. Cardiovascular diseases: the unanswered problem of the century

Introduction

- Line 38: “such as apoptosis or necrosis” – should it be "by apoptosis or ischemic necrosis?"

2. Phospholipids: old knowledge rediscovered

- Lines 71-72: Quote 9: "....such as sphingomyelin (SM), phosphatidylinositol (PI), phosphatidylserine (PS) and phosphatidylglycerol (PG) [9]” but in rat heart studies... Isolated rat heart preparation- Sprague-Dawley rats

- Line 77: "In the lungs, PC makes up to 80% of the lipid" Or should it be written dipalmitoylphosphatidylcholine "Amongst the 90% lipid, the large majority is phospholipid (PL), especially phosphatidylcholine (PC) and more specifically dipalmitoyl phosphatidylcholine (DPPC) (Agudelo et al citation 13)

-                      Lines 81-83: "Moreover, the intraperitoneal administration of PC inhibits bacterial adherence/growth, as well as peritoneal carcinosis, through inhibition of intraperitoneal adhesion of tumor cells [16]. In this sentence may be added the doses used: “A single intraperitoneal dose of approximately 75 mg/kg of phosphatidylcholine, for a 30-min exposure time, … and the elimination of peritoneal carcinosis, through inhibition of intraperitoneal adhesion of tumor cells. The latter effect is achieved by a dose of phospholipids equal to 150 mg/kg…”

-                      Line 93: “In the heart, LPA 92 is involved in the suppression of fibrogenesis, through the activation of LPAR2 [21]” …but quote [21]"In this study, we found that mice also had elevated LPA level in peripheral blood, as well as increased cardiac expression of its receptor LPA2 in the early stages after MI. "

-                      Line 103 and citation [26]: “which can be corrected by PE dietary supplementation” [26] Phosphatidylethanolamine deficiency disrupts α-synuclein homeostasis in yeast and worm models of Parkinson's disease…so supplementation in yeast and worms…

- Line 105, citation [27]: it should be written …differentiation of P19 teratocarcinoma cells into cardiac myocytes.

- Lines 155-116, citation 39: “In order to assess inflammation in experimental ischemic conditions such as chronic limb ischemia and acute myocardial infarction [39]. "MCE molecular imaging with MB-PS was performed 1.5 h, 3.0 h, and 6.0 h after brief (10 min) myocardial ischemia in mice"

-Lines 131-132: " In heart diseases, DAG accumulation has been linked to cardiac hypertrophy, but the underlying mechanisms are still unknown [47]. 132...again animal models according to authors citation 47

- Line 155:  misquote... [73]. “significantly increased in the patients with left ventricle chronic ischemia [73], correlating 155... because it refers to liver disease (Liu, Y.; Saiyan, S.; Men, T.Y.; Gao, H.Y.; Wen, C.; Liu, Y.; Zhou, X.; Wu, C.T.; Wang, L.S.; Cui, C.P. Hepatopoietin Cn reduces 645 ethanol-induced hepatoxicity via sphingosine kinase 1 and sphingosine 1-phosphate receptors. J Pathol 2013

3. Healing after myocardial infarction: the unsolved puzzle of the hear (Ok)

3.1. Phospholipids – the quiet leader behind the doors

- Lines 208-209: “Interestingly, this difference is more pronounced after the ischemia/reperfusion procedure, compared with chronic ischemia [98]. But again an animal model…"Cardiac phospholipidome is altered during ischemia and reperfusion in an ex vivo rat model"

-Lines 216-217: However, more than 3 hours of myocardial ischemia induces ca. ???(what does it mean?)15% depletion of the phospholipid’s concentration [96]

There was several citation of human studies results, as for example about ethanolamine use 128]: "… the concentration of ethanolamine in the blood and breast milk is 2 μM (range 0–12 μM) and 46 μM [1], respectively, ...- Line 250: "Administration of PS-containing liposomes has been shown to protect against type 1 and 280 type 2 diabetes [133]…these approach could be extended to other results obtained in humans.

But again animal experimental/isolated cells  results (describe it): Line 321: [154] "Pigs were kept on a high-fat diet"; Line 328... [159]: "Rats"; Line 334: Binding S1P to S1PR2 mediates recruitment of muse cells into the infarcted 3..."Multilineage-differentiating stress enduring (Muse) cells, pluripotent marker stage-specific embryonic antigen-3+ cells, are nontumorigenic endogenous pluripotent-like stem cells obtainable from various tissues including the bone marrow [160]

3.2. Therapeutical strategies: are they really novel?

- Lines 400-404, citation[175]: “Increased accumulation of PEG–PE micelles in the area of experimental myocardial infarction “…in rabbits; -citation [176]: in the cited article is there some reference to "Phospholipid liposomes ??are used more and more in the last period as delivery systems for different drugs to the infarcted areas, including RNA-based

Lines 426-428: Further, interfering with sphingolipids by various sphingomyelinase inhibitors such as amitriptyline and desipramine (???) is reported to decrease ceramide accumulation induced by ischemia/reperfusion injury and to reduce cardiomyocyte apoptosis [148]… amitriptyline and desipramine (???) in the cited article??

- Again Line 408/citation 131: animal model...." phosphatidylserine oral supplementation in a mouse model of acute myocardial infarction (AMI). We found out that phosphatidylserine increases, significantly, the cardiomyocyte survival by 50% in an acute model of myocardial ischemia-reperfusion. Similar, phosphatidylserine reduced significantly the infarcted size by 30% and improved heart function by 25% in a chronic model of AMI”.

-  Line 416/citation [178]: it should be stated that it was "such as the ischemic myocardium in patients with anteroseptal MI ... by PET"

- - Line 421: "and activation of enzymes [143]"; What enzymes?

- - Line 425, citation [144]: in an animal model of myocardial infarction

- - Line 428, citation [148]..."Male New Zealand White rabbits"; and "Further, interfering with sphingolipids by various sphingomyelinase inhibitors such as amitriptyline and desipramine ??? is reported to decrease ceramide accumulation induced by ischemia/reperfusion injury and to reduce cardiomyocyte apoptosis [148] or " 0.1 mg/kg iv of the sphingomyelinase blocker tricyclodecan 9 y-xanthate?"

- Line 429, citation [180]: animal model... “In vivo, myocardial fibrosis and cardiac dysfunction were induced using transverse aortic constriction (TAC). AST was administered to mice for 12weeks post-surgery”

- Line 430, citation [181:] mice model... and…perhaps it should be said that Astaxanthin (ASX) is a natural carotenoid with strong antioxidant and anti-inflammatory activities.

- Line 423, citation [145]: …Mice were joined in parabiosis

- Lines 440-442: to correct or complement the statement "Used in singular formulation or combined, inclusive combinations with current drugs, phospholipids proved their positive  and valuable effects in the protection of myocardial tissue, inflammation and fibrosis, but also in angiogenesis, coagulation or cardiac regeneration”… at least in animal models and sometimes in Human pathology

Conclusions

- Lines 452-453: The missing or negligible side effects ??? (till now) of phospholipids administration make ..."

-References

- Correct, according to the requirements and appropriate to the topic in publication.

Figure 1:

- Missing caption for acronyms PC, PA/LPA... PE... SM... CL....in alphabetical order

- As in the other boxes, of the Figure 1,  the initial letter should be capitalized... For example, mitochondrial bioenergetics... should be Mitochondrial bioenergetics... Quality---

English correct

Author Response

Comments to the Referee 1:

Despite my opinion transcription regarding what is requested in the electronic form from International Journal of Molecular Sciences MDPI, I would like to add the following comments:

The article is generally good and follows all the precepts required by International Journal of Molecular Sciences MDPI and is therefore in accordance with its publication standards.

The content of the article is interesting, correct, and important in the context of acute myocardial infarction. In fact, it is a scientific area that has been forgotten regarding the area of acute myocardial infarction and the shadows that exist regarding its evolution and the methodology to be used to control and minimize its deleterious effects on the individual who suffers it; The data described point to a window of therapeutic opportunities that this pathology, so frequent and so important, demands of all of us. It is a narrative review with 3 well-structured chapters with scientifically correct content and appropriate citations.

Answer: We thank very much this referee for appreciation of our paper and for the constructive comments. We have now responded and changed the manuscript accordingly.

However, overall it has three aspects that should be improved and that will be mentioned later: there should be an indication throughout the text, whenever possible, of the experimental animal/human species model in which the referenced results were obtained; the plasma concentrations obtained (at least in some studies) of the phospholipids described, and the dose of some phospholipids used orally; incidentally it may be important to construct a Table [as for example with the topics: Reference No. of participants Study design Results (intervention group versus control) P values), or of another type]; there is still one or another inaccuracy to be noted which will be properly referenced below.

Answer: We thank this referee for this suggestion, we have tried now to organize the information in a table (See new table 1 and new table 2) and mention the dosage of phospholipids where possible.

So, and throughout the "article topics" (in red purposeals to be potentially changed)

Title/authors: 3Business Academy Aarhus, Denmark was missing.

Answer: We have now completed the addresses and country in the affiliations.

Abstract

- "This review focus on the current knowledge about the role of phospholipids in the pathophysiology (missing) and therapy of cardiovascular diseases"

 Answer: We have now added this information in the abstract.

  1. Cardiovascular diseases: the unanswered problem of the century

Introduction

 - Line 38: “such as apoptosis or necrosis” – should it be "by apoptosis or ischemic necrosis?"

 Answer: Thank you for pointing this out, We have now reformulated this information as suggested.

  1. Phospholipids: old knowledge rediscovered

- Lines 71-72: Quote 9: "....such as sphingomyelin (SM), phosphatidylinositol (PI), phosphatidylserine (PS) and phosphatidylglycerol (PG) [9]” but in rat heart studies... Isolated rat heart preparation- Sprague-Dawley rats

 Answer: Thank you for pointing this out, We have now added this information in the content.

-Line 77: "In the lungs, PC makes up to 80% of the lipid" Or should it be written dipalmitoylphosphatidylcholine "Amongst the 90% lipid, the large majority is phospholipid (PL), especially phosphatidylcholine (PC) and more specifically dipalmitoyl phosphatidylcholine (DPPC) (Agudelo et al citation 13)

 Answer: We have now changed this information accordingly.

- Lines 81-83: "Moreover, the intraperitoneal administration of PC inhibits bacterial adherence/growth, as well as peritoneal carcinosis, through inhibition of intraperitoneal adhesion of tumor cells [16]. In this sentence may be added the doses used: “A single intraperitoneal dose of approximately 75 mg/kg of phosphatidylcholine, for a 30-min exposure time, … and the elimination of peritoneal carcinosis, through inhibition of intraperitoneal adhesion of tumor cells. The latter effect is achieved by a dose of phospholipids equal to 150 mg/kg…”

Answer: We are very grateful for this observation. We have now added the dosage of PS in the revised manuscript when applicable.

- Line 93: “In the heart, LPA 92 is involved in the suppression of fibrogenesis, through the activation of LPAR2 [21]” …but quote [21]"In this study, we found that mice also had elevated LPA level in peripheral blood, as well as increased cardiac expression of its receptor LPA2 in the early stages after MI. "

Answer: We have now corrected the information in the revised manuscript according to the reference 21.

-  Line 103 and citation [26]: “which can be corrected by PE dietary supplementation” [26] Phosphatidylethanolamine deficiency disrupts α-synuclein homeostasis in yeast and worm models of Parkinson's disease…so supplementation in yeast and worms…

Answer: We have now corrected the information in the revised manuscript according to the reference 26.

- Line 105, citation [27]: it should be written …differentiation of P19 teratocarcinoma cells into cardiac myocytes.

Answer: We have now corrected the information in the revised manuscript according to the reference 27.

- Lines 155-116, citation 39: “In order to assess inflammation in experimental ischemic conditions such as chronic limb ischemia and acute myocardial infarction [39]. "MCE molecular imaging with MB-PS was performed 1.5 h, 3.0 h, and 6.0 h after brief (10 min) myocardial ischemia in mice"

Answer: We have now corrected the information in the revised manuscript according to the reference 39.

-Lines 131-132: " In heart diseases, DAG accumulation has been linked to cardiac hypertrophy, but the underlying mechanisms are still unknown [47]. 132...again animal models according to authors citation 47

Answer: We have now corrected the information in the revised manuscript according to the reference 47.

- Line 155:  misquote... [73]. “significantly increased in the patients with left ventricle chronic ischemia [73], correlating 155... because it refers to liver disease (Liu, Y.; Saiyan, S.; Men, T.Y.; Gao, H.Y.; Wen, C.; Liu, Y.; Zhou, X.; Wu, C.T.; Wang, L.S.; Cui, C.P. Hepatopoietin Cn reduces 645 ethanol-induced hepatoxicity via sphingosine kinase 1 and sphingosine 1-phosphate receptors. J Pathol 2013

Answer: We apologize for this inconvenience; we have now corrected the information in the revised manuscript according to the reference 70.

  1. Healing after myocardial infarction: the unsolved puzzle of the hear (Ok)

3.1. Phospholipids – the quiet leader behind the doors

- Lines 208-209: “Interestingly, this difference is more pronounced after the ischemia/reperfusion procedure, compared with chronic ischemia [98]. But again an animal model…"Cardiac phospholipidome is altered during ischemia and reperfusion in an ex vivo rat model"

Answer: We have now corrected the information in the revised manuscript according to the reference 98 (Current reference 99).

-Lines 216-217: However, more than 3 hours of myocardial ischemia induces ca. ???(what does it mean?)15% depletion of the phospholipid’s concentration [96]

Answer: We have now reformulated the information in the revised manuscript according to the reference 96 (Current reference 97).

There was several citation of human studies results, as for example about ethanolamine use 128]: "… the concentration of ethanolamine in the blood and breast milk is 2 μM (range 0–12 μM) and 46 μM [1], respectively, ...- Line 250: "Administration of PS-containing liposomes has been shown to protect against type 1 and 280 type 2 diabetes [133]…these approach could be extended to other results obtained in humans.

Answer: We have now reformulated the information in the revised manuscript according to the appointed reference.

But again animal experimental/isolated cells results (describe it): Line 321: [154] "Pigs were kept on a high-fat diet"; Line 328... [159]: "Rats"; Line 334: Binding S1P to S1PR2 mediates recruitment of muse cells into the infarcted 3..."Multilineage-differentiating stress enduring (Muse) cells, pluripotent marker stage-specific embryonic antigen-3+ cells, are nontumorigenic endogenous pluripotent-like stem cells obtainable from various tissues including the bone marrow [160]

Answer: We have now reformulated the information in the revised manuscript according to the appointed reference. Now we have specified the experimental model when mentioned in all the revised manuscript.

3.2. Therapeutical strategies: are they really novel?

 - Lines 400-404, citation[175]: “Increased accumulation of PEG–PE micelles in the area of experimental myocardial infarction “…in rabbits; -citation [176]: in the cited article is there some reference to "Phospholipid liposomes ??are used more and more in the last period as delivery systems for different drugs to the infarcted areas, including RNA-based

Answer: We have now reformulated the information in the revised manuscript according to the appointed reference. Now we have specified the experimental model when mentioned in all the revised manuscript.

Lines 426-428: Further, interfering with sphingolipids by various sphingomyelinase inhibitors such as amitriptyline and desipramine (???) is reported to decrease ceramide accumulation induced by ischemia/reperfusion injury and to reduce cardiomyocyte apoptosis [148]… amitriptyline and desipramine (???) in the cited article??

Answer: We apologize for this inconvenience; we have now corrected the information in the revised manuscript according to the reference 148. Now we have added an explanatory information regarding the raised issue and added a novel reference (PMID: 25228885, novel reference 185)

Again Line 408/citation 131: animal model...." phosphatidylserine oral supplementation in a mouse model of acute myocardial infarction (AMI). We found out that phosphatidylserine increases, significantly, the cardiomyocyte survival by 50% in an acute model of myocardial ischemia-reperfusion. Similar, phosphatidylserine reduced significantly the infarcted size by 30% and improved heart function by 25% in a chronic model of AMI”.

Answer: We have now reformulated the information in the revised manuscript according to the correspondent animal model.

-  Line 416/citation [178]: it should be stated that it was "such as the ischemic myocardium in patients with anteroseptal MI ... by PET"

Answer: We have now reformulated the information in the revised manuscript according to the suggestion.

- - Line 421: "and activation of enzymes [143]"; What enzymes?

Answer: We have added “mitochondrial” enzyme, according to the reference 143 (current reference 144).

- - Line 425, citation [144]: in an animal model of myocardial infarction 

Answer: We have now specified the animal model, as suggested.

- - Line 428, citation [148]..."Male New Zealand White rabbits"; and "Further, interfering with sphingolipids by various sphingomyelinase inhibitors such as amitriptyline and desipramine ???is reported to decrease ceramide accumulation induced by ischemia/reperfusion injury and to reduce cardiomyocyte apoptosis [148] or " 0.1 mg/kg iv of the sphingomyelinase blocker tricyclodecan 9 y-xanthate?"

Answer: We have now added the animal model and the dosage of sphingomyelinase inhibitor in the revised manuscript according to the reference 148. Now we have added an explanatory information regarding amitriptyline and desipramine and added a novel reference (PMID: 25228885, novel reference 185).

- Line 429, citation [180]: animal model... “In vivo, myocardial fibrosis and cardiac dysfunction were induced using transverse aortic constriction (TAC). AST was administered to mice for 12weeks post-surgery”

Answer: We have now specified the animal model, as suggested.

- Line 430, citation [181:] mice model... and…perhaps it should be said that Astaxanthin (ASX) is a natural carotenoid with strong antioxidant and anti-inflammatory activities. 

Answer: We have now corrected the information, as suggested.

- Line 423, citation [145]: …Mice were joined in parabiosis

Answer: We have now specified the animal model, as suggested.

- Lines 440-442: to correct or complement the statement "Used in singular formulation or combined, inclusive combinations with current drugs, phospholipids proved their positive and valuable effects in the protection of myocardial tissue, inflammation and fibrosis, but also in angiogenesis, coagulation or cardiac regeneration”… at least in animal models and sometimes in Human pathology

Answer: We have now added the information, as suggested.

Conclusions

- Lines 452-453: The missing or negligible side effects ??? (till now) of phospholipids administration make ..." 

Answer: We thank this referee for this suggestion, we have now added an entire paragraph describing the known side effects of the PL administration. We have now stated that: “Phospholipids have minimal side effects due to their preexistent role as membrane components, thus being a class of compounds already familiar and abundant with the human body. Nevertheless, their oral administration was shown to have some adverse effects, particularly PC. Gut microbiota has the ability to metabolize PC to trimethyla-mine-N-oxide (TMAO), a molecule which has been linked to CVDs and increased mor-tality (especially in the diabetic population). (PMID: 22221489, new reference 189). Initially, the aforemen-tioned health complications could be correlated only with TMAO alone, but not with dietary PC intake. However, a study conducted in the US managed to illustrate a link between dietary consumption of PC and higher probability of CVDs development and mortality (affecting more pronouncedly the diabetic demographic). (PMID: 27281307, new reference 190) Other side effects of PC such as diarrhea, dizziness, nausea and intermenstrual bleeding were reported by patients undergoing subcutaneous injections of PC for the purpose of subcutaneous fat reduction. These symptoms were described by approximately 3% of the studied patient cohort, while other localized, milder symptoms such as pain, swelling, tenderness were reported by a larger number of patients. The study surveying the risks of these procedure concluded that it had minimal risks when performed by experienced doctors, indicating that some of the side effects may have more to do with the method of administration and with the human element. (PMID: 17177743, new reference 191)

In addition, PLs can even reduce the side effects of some drug classes, among which NSAIDs. Administration of 100% native soybean PC for 14 days lessened the gastroin-testinal side-effects of NSAIDS such as gastroduodenal ulcer and abdominal pain. (PMID: 22221489, new reference 189).” (see section 3.2. last paragraph)

-References

- Correct, according to the requirements and appropriate to the topic in publication.

Answer: We have now included additional references, to complete the information included in the revised manuscript.

Figure 1: 

Missing caption for acronyms PC, PA/LPA... PE... SM... CL....in alphabetical order

As in the other boxes, of the Figure 1, the initial letter should be capitalized... For example, mitochondrial bioenergetics... should be Mitochondrial bioenergetics... Quality---

Answer: We have now the information in the figure in the alphabetical order. We have now capitalized all the initial letter, as suggested (see novel Figure Abstract). 

Reviewer 2 Report

Comments:

§  Please check for grammar and any typos once again. Proof-read it multiple times by different people. Just ensure there are no typos and grammatical errors.

§  Line 81-85: Is it possible for a patient with ulcerative colitis to develop or suffer from cardiovascular mortality? If yes, then do you have a way to prevent it or reduce the risk of it? Because in that case the thing (phospholipids) supposed to make you better is killing you rt?

§  Section 2 is information intensive. I think it can be summarized somehow in a table or pictorial format. This is important in reviews to deliver the main message to the readers and have more visual impact.

§  Line 187-199: Author states the neutrophils being a problem in the inflammatory phase. Will there be any chances of cytokine storm developing? If yes, then what are the chances of occurrence, time period, and what will be the repercussions? Cytokine storms are proven to be lethal. If not, explain why.

§  I really liked the way the authors transitioned from section 2 to section 3.

§  Section 3.1 is very elaborate. Please try to summarize with the help of tables etc. All info shouldn’t be included in the table but highlight the main information.

§  Line 353, 395: Every scientific name needs to be italicized.

§  Conclusion should include any new perspectives from authors. Given all the literature review, what do the authors think will be a pivoting decision to develop phospholipids as treatment? Which phospholipids will likely be target, why and how the experiments would go with? Any review is incomplete without concrete future steps. I don’t expect the authors to have detailed experimental design but just some indication.

§  Figure 1: Needs a key for all those cell types and red circles. It’s hard to decipher it.

§  Lastly, I would like to appreciate the efforts of the authors in drafting a very well organized and comprehensive study here.

Author Response

Comments to the Referee 2:

Please check for grammar and any typos once again. Proof-read it multiple times by different people. Just ensure there are no typos and grammatical errors.

Answer: We have proved and read the manuscript for any typos or grammatical errors.

  • Line 81-85: Is it possible for a patient with ulcerative colitis to develop or suffer from cardiovascular mortality? If yes, then do you have a way to prevent it or reduce the risk of it? Because in that case the thing (phospholipids) supposed to make you better is killing you rt?

Answer: We are grateful this referee for pointing this out, indeed, we have found that ulcerative colitis is considered a risk factor for cardiovascular disease and the treatment for it can reduce the incidence of the cardiovascular events. We have stated this now in our revised manuscript: “Interestingly, since ulcerative colitis is considered a risk factor linked to cardiovascular diseases such as accelerated atherosclerosis, atrial fibrillation and heart failure [PMID: 31653485 – currently reference 172, PMID: 36058305 – currently reference 173], using PC enriched enteric lecithin to improve remission rates [PMID: 35831742– currently reference 174] can reduce corticosteroid dependence [PMID: 17975182– currently reference 175] and thus, reduce stress associated cardiovascular events” (see section 3.2. paragraph 6).

  • Section 2 is information intensive. I think it can be summarized somehow in a table or pictorial format. This is important in reviews to deliver the main message to the readers and have more visual impact.

Answer: We thank very much for pointing this out. We have now included a comprehensive table to summarize the information included in this chapter (see novel table 1).

  • Line 187-199: Author states the neutrophils being a problem in the inflammatory phase. Will there be any chances of cytokine storm developing? If yes, then what are the chances of occurrence, time period, and what will be the repercussions? Cytokine storms are proven to be lethal. If not, explain why.

Answer: We agree with the referee that this is an important process taken place immediately after myocardial infarction, therefore we have now mentioned the “cytokines storm” as an important event induced by the neutrophils. We have now stated that: “Neutrophil infiltration induces shortly after myocardial infarction a massive upregulation of all kinds of cytokines and chemokines, so called “cytokine storm”, which correlate directly with the size of the affected areas, leading to additional cardiomyocyte apoptosis, thus worsening prognosis” (PMID: 35876879, new reference 93) (see section 3, second paragraph).  

  • I really liked the way the authors transitioned from section 2 to section 3. Section 3.1 is very elaborate. Please try to summarize with the help of tables etc. All info shouldn’t be included in the table but highlight the main information.

Answer: We have now included a comprehensive table to summarize the information included in this chapter (see novel table 2). We also added a novel Figure 2 to summarize the different therapeutical strategies possible during healing after myocardial infarction using phospholipids as target (see novel Figure 2)

  • Line 353, 395: Every scientific name needs to be italicized.

Answer: We are grateful for pointing this out. We have italicized all the scientific names.

  • Conclusion should include any new perspectives from authors. Given all the literature review, what do the authors think will be a pivoting decision to develop phospholipids as treatment? Which phospholipids will likely be target, why and how the experiments would go with? Any review is incomplete without concrete future steps. I don’t expect the authors to have detailed experimental design but just some indication.

Answer: We appreciate this suggestion very much. We have now included an all paragraph about the future perspectives in using phospholipids as therapeutic agents (see novel section 3.3. Future perspective: simple is complicated)

  • Figure 1: Needs a key for all those cell types and red circles. It’s hard to decipher it.

Answer: We thank very much for pointing this out. We have now included a comprehensive description for the graphic elements in the Figure 1 (see novel Figure Abstract).

  • Lastly, I would like to appreciate the efforts of the authors in drafting a very well organized and comprehensive study here.

Answer: We thank very much this referee for the appreciation of our manuscript and for the constructive comments and suggestions.
